# Biological Breakthroughs and Drug Discovery Revolution via Cryo-Electron Microscopy of Membrane Proteins

**DOI:** 10.3390/membranes15120368

**Published:** 2025-12-01

**Authors:** Vitor Hugo Balasco Serrão

**Affiliations:** 1Biomolecular Cryo-EM Facility, University of California—Santa Cruz, Santa Cruz, CA 95064, USA; vbalasco@ucsc.edu; 2Chemistry and Biochemistry Department, University of California—Santa Cruz, Santa Cruz, CA 95064, USA

**Keywords:** Cryo-EM, structure-based drug discovery, membrane proteins, structural biology

## Abstract

The application of cryo-electron microscopy (cryo-EM) in membrane protein structural biology has catalyzed unprecedented advances in our understanding of fundamental biological processes and transformed drug discovery paradigms. This review briefly describes the biological achievements enabled using cryo-EM techniques, including single particle analysis (SPA), micro-electron diffraction (microED), and subtomogram averaging (STA), in elucidating the structures and functions of membrane proteins, ion channels, transporters, and viral glycoproteins. We highlight how these structural insights have revealed druggable sites, enabled structure-based drug design, and provided mechanistic understanding of disease processes. Key biological targets include G protein-coupled receptors (GPCRs), ion channels implicated in neurological disorders, respiratory chain complexes, viral entry machinery, and membrane transporters. The integration of cryo-EM with computational drug design has already yielded clinical candidates and approved therapeutics, marking a new era in membrane protein pharmacology.

## 1. Introduction: The Membrane Protein Drug Discovery Challenge

Biological membranes represent one of the most fundamental organizational principles in living systems, serving as selective barriers that compartmentalize cellular processes while facilitating essential transport and signaling functions [1]. The lipid bilayer structure, first proposed by Gorter and Grendel in 1925 [2] and later refined in the fluid mosaic model of Singer and Nicolson, provides the architectural foundation for membrane proteins that comprise approximately 20–30% of all proteins in living organisms and represents the target for over 60% of currently marketed drugs [3]. Despite their biological and pharmaceutical importance, membrane proteins have historically been underrepresented in structural databases due to inherent challenges in their purification, crystallization, and structure determination [4,5]. This apparent paradox reflects both the therapeutic importance of membrane proteins and historical challenges in obtaining their high-resolution structures. Ion channels regulate neuronal signaling and cardiac function, G protein-coupled receptors (GPCRs) mediate hormonal and sensory responses, transporters maintain cellular homeostasis, and viral membrane proteins facilitate pathogen entry and replication. Traditional X-ray crystallography, while successful for many soluble proteins, faces significant obstacles when applied to membrane proteins, including difficulties in obtaining well-diffracting crystals and the artificial environment imposed by crystal contacts.

The advent of cryo-electron microscopy (cryo-EM) has fundamentally transformed membrane protein structural biology (Figure 1) [6]. Unlike X-ray crystallography, cryo-EM enables the visualization of proteins in near-native states without the need for crystallization, making it particularly well-suited for studying flexible, dynamic, and large membrane protein complexes. The method has evolved rapidly, with improvements in detector technology, image processing algorithms, and computational methods leading to what is often termed the “resolution revolution” in cryo-EM [6,7,8].

The cryo-EM revolution has fundamentally transformed this landscape. The ability to determine near-atomic resolution structures of membrane proteins in native-like environments has unveiled druggable sites previously inaccessible to structure-based drug design. More importantly, cryo-EM has revealed the dynamic nature of membrane proteins, capturing multiple conformational states that reflect functional mechanisms and providing insights into allosteric regulation, cooperativity, and drug action [9,10]. This review examines the biological achievements enabled by cryo-EM techniques (Figure 2 and summarized in Table 1) applied to membrane systems, emphasizing how structural insights have translated into therapeutic opportunities and advanced our understanding of human disease.

## 2. Technical Foundations of Cryo-EM for Membrane Studies

### 2.1. Single Particle Analysis (SPA)

Single particle analysis cryo-EM has become the gold standard for determining the high-resolution structures of membrane proteins. The technique involves embedding purified membrane protein samples in vitreous ice, imaging thousands to millions of individual particles, and computationally combining these images to reconstruct three-dimensional density maps [11].

SPA represents the most widely used cryo-EM approach for high-resolution structure determination of purified membrane proteins (Figure 3) [12]. In SPA, identical or similar particles are imaged in vitreous ice and computationally aligned and averaged to improve the signal-to-noise ratio and achieve high resolution [13]. For membrane proteins, SPA typically involves solubilization in detergents, lipid nanodiscs, or amphipols to maintain protein stability while removing the native membrane environment.

The success of cryo-EM in membrane protein studies has been facilitated by several key methodological developments. Membrane scaffold proteins (MSPs) can be used to reconstitute membrane proteins into lipid bilayer patches surrounded by apolipoprotein A-I-derived proteins. This approach more closely mimics the native membrane environment and has proven particularly valuable for conformationally dynamic membrane proteins [14]. Lipid nanodiscs have emerged as particularly important tools for membrane protein reconstitution, providing a more native-like environment compared to detergent micelles while maintaining the discrete particles required for SPA [15]. These disc-shaped lipid bilayers, stabilized by membrane scaffold proteins, allow membrane proteins to be studied in defined lipid environments. Traditional approaches use mild detergents such as lauryl maltose neopentyl glycol (LMNG), n-dodecyl-β-D-maltoside (DDM), or digitonin to extract membrane proteins while preserving their native folds. The choice of detergent significantly impacts particle behavior on cryo-EM grids, with some detergents promoting preferred orientations or causing protein aggregation. Detergent screening and optimization remain crucial for successful membrane protein cryo-EM studies. Mild detergents such as LMNG and glyco-diosgenin (GDN) have proven particularly effective for maintaining protein stability and generating high-quality cryo-EM samples [16]. A newer approach involves reconstituting membrane proteins into small lipid vesicles or bicelles, providing a fully bilayer environment while maintaining particle monodispersity suitable for SPA [17], including the design of peptidiscs [18,19].

Finally, image processing advances have been equally important. Membrane proteins present unique challenges for cryo-EM image processing. The presence of detergent micelles or lipid bilayers creates additional density that must be properly accounted for during particle picking, 2D classification, and 3D reconstruction. The development of sophisticated algorithms for particle picking, classification, and refinement has enabled the determination of structures from increasingly challenging samples. Automated particle picking algorithms must distinguish membrane protein particles from background noise, ice contamination, and empty micelles or nanodiscs. Template-based picking using low-pass filtered projections of initial models has proven effective, though care must be taken to avoid bias toward specific orientations. In addition, recent advances in local resolution refinement and multi-body refinement have been particularly beneficial for membrane proteins, which often exhibit flexible domains and conformational heterogeneity [20,21]. The implementation of Bayesian approaches in software packages such as RELION [22], cryoSPARC [20], and cisTEM [23] has dramatically improved the quality of membrane protein reconstructions. These methods can account for continuous conformational variability and have enabled the resolution of previously intractable membrane protein structures

### 2.2. Subtomogram Averaging (STA)

Subtomogram averaging involves collecting tomographic tilt series of frozen–hydrated samples, reconstructing three-dimensional tomograms, and then identifying, extracting, and averaging subvolumes containing the structure of interest [24]. This approach is particularly powerful for studying membrane proteins in their native membrane context or in reconstituted systems that closely mimic biological membranes.

STA represents a powerful approach for studying membrane proteins within their native cellular context. Unlike SPA, which requires purified proteins, STA can determine the structures of membrane proteins directly within cells, organelles, or native membranes.

Specimens such as cells, organelles, or membrane vesicles are vitrified either by plunge-freezing (for thin samples) or high-pressure freezing followed by focused ion beam (FIB) milling to create lamellae thin enough for electron transparency [25,26]. The sample is imaged at multiple tilt angles (typically ±60°) to collect a tomographic tilt series. Missing wedge artifacts due to limited tilting range represent a fundamental limitation of the technique [24,27]. Individual tilt series are computationally combined to generate three-dimensional tomograms showing the cellular ultrastructure in molecular detail. Data can be interpreted using cryo-ET by segmenting regions of interest such as organelles, membranes, particles and cellular compartments, reflecting ~5.3% of the observable data in the database (Figure 3). Particularly for membrane proteins of interest, identification within tomograms is performed either manually or through template matching approaches and small sub-volumes (subtomograms) containing individual particles are extracted [28]. The extracted subtomograms are aligned and averaged to generate high-resolution structures while preserving information about the native membrane environment and protein–protein interactions [24]. Despite its power, STA faces several inherent limitations. One of the most important is the anisotropic resolution due to the missing wedge artifact, with the worst resolution typically along the beam direction [24].

Recent technical developments have dramatically improved the resolution and applicability of STA. Cryo-focused ion beam milling, or cryo-FIB-SEM, enables the preparation of thin lamellae from virtually any cellular sample, expanding STA applications to thick specimens, including tissue samples and large cells [29]. The integration of STA with fluorescence microscopy, X-ray tomography, and other imaging modalities provides complementary information about protein localization, dynamics, and function, which includes the possibility of using correlative light-electron microscope in order to identify the target of interest with precision [26,30]. Even with FIB milling, sample thickness limits resolution, and thick samples suffer from multiple scattering effects.

Tomograms have very weak signals due to the low dose used during data acquisition in order to mitigate radiation damage on the sample. Lately, the development of Volta [31] and laser [32] phase plates has improved the contrast of weakly scattering biological specimens, which is particularly important for membrane proteins where contrast can be limited.

Similar to the newest advances in SPA data processing, developments in subtomogram averaging algorithms, including the implementation of multi-particle refinement and continuous conformational variability analysis, have pushed resolution limits toward the atomic scale [33], including the possibility of studying conformational heterogeneity *in situ* [34]. However, excessive conformational variability limits the resolution achievable through averaging.

### 2.3. Micro-Electron Diffraction (microED)

Micro-electron diffraction represents a revolutionary approach that combines the principles of X-ray crystallography with the advantages of electron microscopy [35]. Traditional X-ray crystallography often requires membrane protein crystals of 50–100 μm or larger, which can be extremely difficult to obtain due to the challenging nature of membrane protein crystallization. MicroED enables structure determination from crystals that are 1000-fold smaller in volume. However, some limitations are related to the fact that membrane protein crystals suitable for microED must still diffract well and have low mosaicity. The incorporation of detergents or lipids can sometimes compromise crystal quality.

Protein crystals are prepared using conventional crystallization methods but are allowed to form much smaller crystals than would typically be useful for X-ray diffraction. For membrane proteins, crystallization in lipidic cubic phases (LCPs) or traditional detergent-based conditions can be employed [36]. Crystals are vitrified in liquid ethane or propane, similar to standard cryo-EM sample preparation, preserving their native hydration state and preventing radiation damage during data collection. The sample is illuminated with a parallel electron beam, and diffraction patterns are collected while continuously rotating the crystal. Modern direct electron detectors enable the collection of high-quality diffraction data with minimal radiation damage [36]. While both X-rays and electrons cause radiation damage, the physics of electron scattering allow for efficient data collection at cryogenic temperatures where damage rates are minimized. The shorter wavelength of electrons also provides better phase information than X-rays. The rapid data collection possible with microED opens up possibilities for time-resolved structural studies of membrane proteins, potentially capturing intermediate states in transport or catalytic cycles [37].

Diffraction images are processed using modified X-ray crystallography software packages, with corrections applied for the unique properties of electron diffraction, including dynamic scattering effects and the relationship between structure factors and electron scattering. Like X-ray crystallography, microED faces the phase problem, requiring molecular replacement, heavy atom derivatives, or other phasing approaches [38].

## 3. Revolutionary Impact on Membrane Protein Structure Determination

### 3.1. Breakthrough Structures

The impact of cryo-EM on membrane protein structural biology has been profound and transformative. Several landmark studies have demonstrated the technique’s power in revealing previously inaccessible structures, as summarized in Figure 4.

The structure of TRPV1 (transient receptor potential vanilloid 1—EMD-8118/PDB ID 5IRZ) ion channel determined using SPA marked a significant milestone, providing insights into channel gating mechanisms at near-atomic resolution [39]. This work demonstrated that cryo-EM could achieve resolution comparable to X-ray crystallography for membrane proteins while preserving physiologically relevant conformations.

Followed by ion channels, voltage-gated ion channels have been particularly well-served by cryo-EM approaches. The structures of voltage-gated sodium channels (e.g., rabbit ryanodine receptor 1—EMD-38044/PDB ID8X4A 8 × 4A) [40] and calcium channels [41], revealed detailed mechanisms of voltage sensing and ion selectivity that were previously inaccessible. These studies highlighted cryo-EM’s ability to capture different conformational states, providing dynamic pictures of channel function.

Another remarkable achievement was the γ-secretase complex (EMD-3238/PDB ID 5FN3) [42]. This multi-subunit membrane protease, responsible for processing amyloid precursor protein and linked to Alzheimer’s disease, was structurally characterized through SPA. The structures revealed the mechanism of substrate recognition and cleavage within the lipid bilayer, providing insights into disease-related mutations and potential therapeutic targets.

Large respiratory complexes from both prokaryotic and eukaryotic sources have been extensively studied using SPA with countless structures (for example, swine ATP synthase—EMD-45012/PDB ID 9BXU) [43,44,45,46,47,48]. The structures of Complexes I, III, and IV have been determined at near-atomic resolution, revealing electron transfer pathways, proton pumping mechanisms, and the role of lipid cofactors in respiratory chain function.

Along the same lines, neurotransmitter receptors such as GABAA receptor (EMD-45878/PDB ID 9CRS) [49], NMDA receptor [50], and other ligand-gated ion channels have been structurally characterized in multiple functional states through SPA [51,52,53]. These studies have revealed allosteric mechanisms, drug binding sites, and the structural basis of channel gating and desensitization.

Membrane fusion is a fundamental cellular process essential for vesicle trafficking, neurotransmitter release, and viral infection [54]. SNARE (Soluble N-ethylmaleimide-sensitive factor Attachment protein REceptor) proteins are the core machinery driving membrane fusion in eukaryotic cells and cryo-EM has provided unprecedented insights into SNARE-mediated fusion mechanisms. Studies using cryo-electron tomography have revealed the organization of SNARE proteins during different stages of fusion. The visualization of SNARE complexes in reconstituted fusion systems has shown how these proteins facilitate membrane merger and the role of complexin and other regulatory proteins in controlling fusion kinetics [55,56]. These studies have been crucial in understanding synaptic vesicle fusion and its regulation.

Other relevant biological processes are mitochondrial membrane fusion and fission, which are essential for organelle maintenance and cellular energy metabolism. Cryo-EM studies of mitofusins and other proteins involved in mitochondrial dynamics have revealed the molecular mechanisms governing these processes [57]. These studies have shown how conformational changes in fusion proteins drive membrane merger and how mutations in these proteins lead to neurodegenerative diseases.

Finally, it is imperative to mention the importance of ABC transporters. ATP-binding cassette (ABC) transporters represent one of the largest and most functionally diverse protein superfamilies, which are responsible for the active translocation of a wide variety of substrates, including ions, lipids, peptides, drugs, and metabolic products, across cellular membranes (e.g., human ABCA4—EMD-23617/7M1P) [58]. These transporters harness the energy derived from ATP binding and hydrolysis to drive conformational changes that alternate the accessibility of the substrate-binding site between inward- and outward-facing states, thereby ensuring unidirectional transport. Structurally, ABC transporters are typically composed of two transmembrane domains (TMDs), which form the substrate translocation pathway, and two cytosolic nucleotide-binding domains (NBDs), where ATP hydrolysis occurs, similarly to GPCRs, which have become one of the holy grails for druggability in structural biology.

The advent of SPA has revolutionized the understanding of ABC transporter mechanisms by enabling high-resolution visualization of these dynamic molecular machines in multiple functional states without the need for crystallization. Recent cryo-EM studies have captured the distinct conformational intermediates of transporters such as P-glycoprotein (ABCB1) [59], the cystic fibrosis transmembrane conductance regulator (CFTR/ABCC7), and bacterial homologs such as MsbA, elucidating how ATP binding induces NBD dimerization and TMD rearrangement to achieve substrate extrusion. Cryo-EM has also provided unprecedented insights into lipid–protein and drug–protein interactions, revealing, for example, how lipid molecules stabilize specific conformations or how inhibitors lock transporters in defined states, offering direct implications for pharmacological targeting. In the case of CFTR, cryo-EM has elucidated the molecular basis of gating defects caused by disease-associated mutations and informed the rational design of corrector and potentiator drugs for cystic fibrosis [60]. Collectively, cryo-EM has transformed ABC transporter research by enabling a near-continuum view of their conformational cycle at near-atomic resolution, uncovering mechanistic principles that were previously inaccessible via X-ray crystallography or other structural methods.

### 3.2. G Protein-Coupled Receptors: The Holy Grail of Drug Targets

SPA has become central to structure-guided drug design because it provides near-atomic resolution structures of purified macromolecules, including membrane proteins and other targets that are difficult or impossible to crystallize. Pharmacologically, SPA provides sufficient detail to resolve ligand poses, hydration networks, and subtle chemical interactions that drive drug affinity and specificity. Its ability to capture multiple functional states is crucial for the design of allosteric modulators, biased agonists, and conformation-selective therapeutics. Importantly, the integration of SPA-derived structures with computational drug design has already yielded clinical candidates and approved therapeutics, marking a new era in membrane protein pharmacology and expanding the set of viable targets for precision drug development.

GPCRs constitute the largest family of membrane receptors, with over 800 members in the human genome mediating responses to light, odors, hormones, and neurotransmitters [61]. They represent the most successful class of drug targets, with GPCR-targeting drugs generating over $180 billion in annual pharmaceutical sales. However, the structural characterization of GPCRs remained elusive for decades owing to their inherent flexibility and low expression levels [61]. G protein-coupled receptors (GPCRs), the largest class of membrane proteins and major drug targets, have also benefited enormously from cryo-EM advances and comprise close to the top-two structures solved using cryo-EM (Figure 5) [62,63,64,65,66].

The breakthrough came with cryo-EM structures of GPCRs in complex with G proteins [67], revealing active-state conformations that were previously inaccessible. These structures have revolutionized our understanding of GPCR activation, signaling bias, and allosteric regulation (for example, human PGD2-bound prostaglandin D2 receptor (DP1)-Gs complex—EMD-47802/PDB ID 9E9S, represented in Figure 4).

GPCRs constitute one of the largest and most pharmacologically relevant families of membrane proteins, playing a pivotal role in transducing extracellular signals into intracellular responses that regulate virtually all physiological processes. Their accessibility at the cell surface, coupled with their diverse signaling mechanisms, makes them exceptionally attractive targets for therapeutic intervention. Indeed, approximately one-third of all approved drugs act by modulating GPCR activity, addressing a wide range of conditions from cardiovascular and neurological disorders to cancer and metabolic diseases. Continued advances in structural biology, particularly cryo-electron microscopy, have provided unprecedented insights into GPCR conformational dynamics, ligand specificity, and signaling bias, paving the way for rational drug design and the development of more selective and efficacious therapeutics

### 3.3. In Situ Visualization of Membranes and TM Proteins in Action

Subtomogram averaging has provided unique insights into how membrane proteins organize within native membrane environments. Studies of respiratory complexes in mitochondrial membranes have revealed supramolecular organization and the role of lipid environment in complex stability. Some examples are summarized in Figure 4 [68,69,70]. This native-state visualization is equally impactful for pharmacology: cryo-ET enables in situ target validation by showing where drug targets are localized, how abundant they are, and how their structural states are modulated by cellular conditions. It can also reveal drug-induced changes in cellular architecture, directly visualizing mechanisms of action and uncovering potential off-target effects or sources of toxicity. By bridging atomic-scale structural information with cell-level physiology, cryo-ET enhances mechanistic pharmacology and strengthens the biological relevance of structure-based drug design. Many medically important targets adopt distinct conformations or assembly states in vivo that differ from those captured in purified samples. STA makes it possible to characterize these native states, revealing disease-relevant conformations and endogenous regulatory interactions that are critical for therapeutic development. From a pharmacological perspective, STA enables researchers to observe how inhibitors or mutations alter the structure and arrangement of macromolecular machines within the cell, providing mechanistic insights that inform the design of next-generation therapeutics.

Some of the remarkable structures solved using STA are respiratory supercomplexes [71,72,73]. Studies of mitochondrial cristae have revealed the organization of respiratory chain complexes into supercomplexes, showing how Complexes I, III, and IV associate to form higher order structures that facilitate efficient electron transfer and minimize reactive oxygen species production. Similarly, the organization of photosystem complexes within thylakoid membranes has been characterized using STA, revealing how light-harvesting complexes are arranged to optimize energy transfer and how the apparatus responds to changing light conditions [74,75,76].

Virus glycoproteins have been commonly observed using X-ray crystallography [77,78,79] and SPA [80,81,82,83,84], but from isolated samples that were previously purified. Currently, STA has been instrumental in characterizing viral glycoproteins in their native membrane environment [56,85,86]. Studies of influenza hemagglutinin [87], HIV envelope proteins [88], and SARS-CoV-2 spike proteins (e.g., SARS-CoV-2 post-fusion spike—EMD-31037/PDB ID 7E9T) [89,90,91,92] have revealed conformational states and membrane interactions that are not accessible through SPA of purified proteins. Enveloped viruses must fuse their lipid envelope with host cell membranes to deliver their genetic material. STA has been instrumental in elucidating viral fusion mechanisms, particularly for influenza virus and HIV. The influenza hemagglutinin (HA) protein undergoes dramatic conformational changes during membrane fusion, and cryo-EM studies have captured key intermediates in this process, hardily trapped using different methods such as X-ray crystallography. These studies have revealed how pH-induced conformational changes in HA lead to membrane fusion and have informed antiviral drug development. Another case study is the HIV envelope protein, and studies using cryo-EM have provided detailed views of the prefusion state and conformational changes triggered by receptor binding [88]. These insights have been crucial for HIV vaccine development and understanding viral entry mechanisms.

The nuclear pore complex (NPC) represents one of the most successful applications of subtomogram averaging. The NPC’s large size, membrane integration, and eight-fold symmetry make it an ideal target for STA approaches. Studies have revealed the detailed architecture of NPCs in different organisms and provided insights into nuclear transport mechanisms (human dilated nuclear pore complex—EMD-14321/PDB ID 7R5J) [93]. Although they technically span the nuclear envelope, nuclear pore complexes represent one of the most successful applications of STA to large membrane-embedded complexes, achieving near-atomic resolution and revealing the mechanism of selective transport.

As the final example of the crucial implementation of STA and its importance, bacterial cell envelopes contain numerous membrane proteins that are essential for survival and pathogenesis. STA studies have revealed the organization of secretion systems, flagellar motors, and other membrane-embedded complexes in their native context [94]. These studies have provided insights into bacterial physiology and identified potential targets for antimicrobial development.

### 3.4. MicroED as a Powerful Method to Understand TM-Proteins

Obtaining high-quality crystals of membrane proteins remains one of the most challenging tasks in structural biology due to their intrinsic biochemical and biophysical properties. Unlike soluble proteins, membrane proteins are embedded within lipid bilayers and rely on this environment to maintain their native conformation and stability. When extracted from membranes, they often lose their structural integrity, leading to aggregation or denaturation. The need to solubilize them using detergents or other membrane-mimetic systems further complicates the process, as finding suitable conditions that preserve both stability and crystallizability is extremely difficult. Additionally, the inherent flexibility and heterogeneity of many membrane proteins, coupled with their low natural abundance and challenges in overexpression, hinder the production of homogeneous and concentrated samples necessary for crystallization. As a result, despite their biological importance, only a small fraction of known membrane proteins has been successfully crystallized and structurally characterized. Several membrane protein structures have been successfully determined using microED.

Pharmacologically, microED plays a vital role in medicinal chemistry by enabling the rapid determination of absolute stereochemistry, polymorphism, and conformational preferences of drug candidates. It also supports fragment-based drug design and offers an orthogonal solution when traditional crystallographic approaches fail, accelerating structure-driven optimization and validating chemical hypotheses with atomic precision.

Bacteriorhodopsin is one example of the structures obtained using microED [95]. This light-driven proton pump was among the first membrane proteins studied using microED, demonstrating the feasibility of the approach for seven-transmembrane proteins. The structure revealed details of the retinal binding site and proton transfer pathway that were consistent with previous X-ray structures but obtained from much smaller crystals.

Another example is some isolated domains from larger ion channel complexes that have been studied using microED (human A2A BRIL adenosine receptor—EMD-29586/PDB ID 8FYN and the murine voltage-dependent anion channel mutant—EMD-23037/PDB ID 7KUH) [96], providing complementary structural information to full-length structures obtained using SPA. Various small membrane proteins, including antimicrobial peptides and membrane-spanning domains, have been characterized using microED [35,97]. These studies have been particularly valuable for understanding the structural basis of membrane disruption and antimicrobial activity.

## 4. Current Challenges and Technical Limitations

Despite remarkable advances, cryo-EM still faces resolution limitations, particularly for smaller membrane proteins. Proteins smaller than ~50 kDa remain challenging targets for high-resolution SPA, although recent advances in detector technology and image processing are gradually lowering this threshold. New advances in detectors and imaging methods (e.g., laser phase plates) have been developed, which should start pushing further the resolution of small molecular weight macromolecules. In addition, new data processing methods and workflows can also help resolve <50 kDa complexes in the near future [98]. Moreover, the choice of membrane mimetics can significantly influence protein structure and dynamics. While lipid nanodiscs provide more native-like environments than detergent micelles, they may still not fully recapitulate the complexity of biological membranes. Developing better membrane mimetics remains an active area of research.

New advances in the development of time-resolved cryo-EM approaches promises to provide dynamic views of membrane protein function and membrane fusion processes. Recent advances in rapid mixing and plunge-freezing techniques are enabling the capture of transient intermediates in membrane protein conformational changes [99].

For in situ analysis, new advances in cryo-focused ion beam milling and subtomogram averaging are enabling structural studies of membrane proteins within intact cells. These in situ approaches promise to reveal how membrane proteins function in their native cellular context, including interactions with other cellular components and responses to physiological stimuli [100]. The combination of cryo-EM with other structural techniques, including X-ray crystallography, NMR spectroscopy, and mass spectrometry, is providing a more complete picture of membrane protein structure and dynamics. These integrative approaches are particularly powerful for understanding large, multi-domain membrane proteins and their regulatory mechanisms.

Finally, cryo-EM is increasingly being used in pharmaceutical research for structure-based drug design targeting membrane proteins. The ability to visualize drug binding sites in near-native conformations is providing new opportunities for developing more effective therapeutics [101].

## 5. Conclusions and Future Outlook

The application of cryo-EM to membrane protein structural biology has ushered in a new era of drug discovery and biological understanding. The technique has solved longstanding questions about membrane protein function, revealed druggable sites that were previously inaccessible via structure-based design, and enabled the development of breakthrough therapeutics for previously intractable diseases. The biological impact extends far beyond individual protein structures. Cryo-EM has revealed the dynamic nature of membrane proteins, showing how they sample multiple conformational states and interact with their lipid environment. This understanding has transformed our view of membrane protein function and opened new avenues for therapeutic intervention.

The integration of cryo-EM with advanced computational drug design is already transforming therapeutic discovery, particularly for membrane proteins that historically resisted structural characterization. By providing high-resolution structural frameworks and capturing multiple physiologically relevant conformations, cryo-EM enables computational approaches such as molecular docking, free-energy calculations, and machine learning-based ligand optimization, to operate with unprecedented accuracy. This synergy has already produced clinical candidates and even approved therapeutics, validating the power of combining experimental structural insight with in silico design. Together, these advances signal a new era in membrane protein pharmacology, one in which cryo-EM–driven structural knowledge directly accelerates the development of more selective, potent, and mechanism-based therapeutics.

The success stories highlighted in this review demonstrate that cryo-EM has matured from a promising technique to an essential tool for membrane protein drug discovery. As technical capabilities continue to advance and our understanding of membrane protein biology deepens, we can anticipate continued breakthroughs that will transform treatment of human disease. The revolution in membrane protein structural biology enabled by cryo-EM represents one of the most significant advances in modern biomedical research. By providing unprecedented insights into the molecular basis of life’s most important biological processes, cryo-EM continues to drive innovations that improve human health and advance our fundamental understanding of biology.

## Figures and Tables

**Figure 1 membranes-15-00368-f001:**
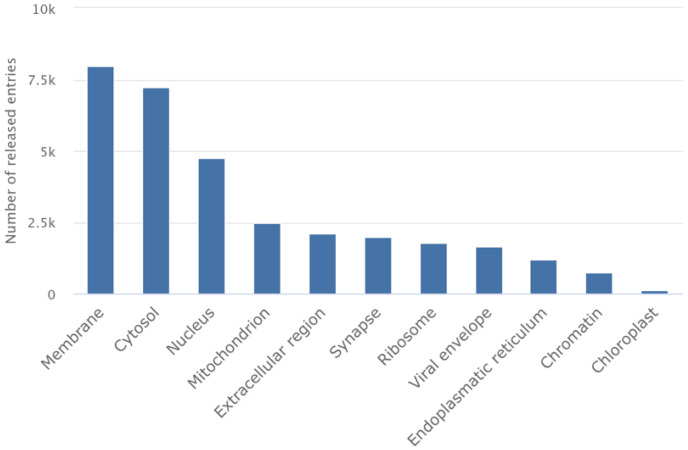
Overall entries in the Electron Microscopy Data Bank (EMDB). Number of entries based on cellular location, revealing that membrane proteins comprise 15.7% of entries (7995 unique entries of 50,997 entries—accessed on November 2025), representing the main application of cryo-EM.

**Figure 2 membranes-15-00368-f002:**
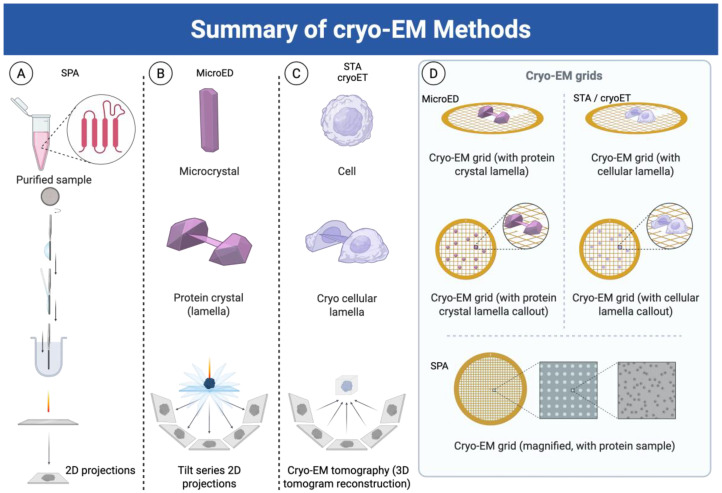
Overview of cryo-electron microscopy (cryo-EM) imaging modalities: single particle analysis (SPA), subtomogram averaging (STA), and microcrystal electron diffraction (microED). Cryo-EM encompasses a suite of imaging techniques that enable the structural determination of biological macromolecules across different sample types and resolutions. (**A**) In SPA, thousands to millions of individual particle projections of purified macromolecules are computationally aligned and averaged to generate high-resolution 3D reconstructions of isolated complexes. (**B**) MicroED captures diffraction patterns from submicron-sized 3D crystals, enabling atomic-resolution structure determination from crystals too small for traditional X-ray crystallography. (**C**) STA combines cryo-electron tomography with averaging of repeated structures within tomograms, allowing visualization of macromolecular assemblies in their native cellular or membrane environments. Together, these complementary cryo-EM methods provide a comprehensive framework for structural biology—bridging molecular to cellular scales and static to dynamic systems. (**D**) Schematic of samples on grids after preparation prior to data acquisition. Created using BioRender.

**Figure 3 membranes-15-00368-f003:**
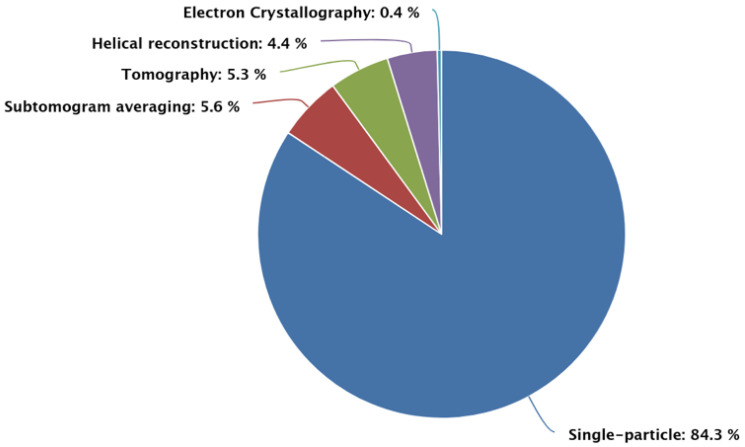
Distribution based on imaging method of deposited structures in EMDB. Overall numbers of EMDB entries based on imaging method—accessed on November 2025.

**Figure 4 membranes-15-00368-f004:**
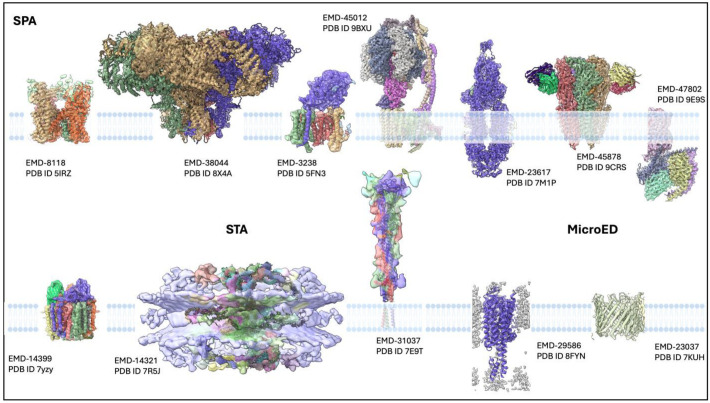
Representative cryo-EM structures of membrane proteins determined using different cryo-EM modalities. Cryo-electron microscopy enables structural determination of membrane proteins across multiple resolution regimes and sample contexts. (Top) Examples of high-resolution single-particle (SPA) reconstruction: EMD-8118 (mouse TRPV1 structures in nanodiscs), EMD-38044 (rabbit Ryanodine receptor 1), EMD-3238 (human gamma secretase), EMD-45012 (swine ATP synthase), EMD-23617 (human ABCA4), EMD-45878 (human GABAA receptor), and EMD-47802 (human PGD2-bound prostaglandin D2 receptor (DP1)-Gs complex). (Bottom) Examples of subtomogram average (STA) of a membrane protein complex embedded within its native lipid bilayer reveals structural organization in situ, preserving macromolecular context and cellular architecture: EMD-14399 (bacterial pMMO), EMD-14321 (human dilated nuclear pore complex), and EMD-31037 (SARS-CoV-2 post-fusion spike). (Bottom) Examples of microcrystal electron diffraction (microED) structure of a membrane protein microcrystal demonstrates the method’s ability to derive atomic-resolution information from nanocrystals that are too small for conventional X-ray crystallography: EMD-29586 (human A2A BRIL adenosine receptor) and EMD-23037 (murine voltage-dependent anion channel mutant).

**Figure 5 membranes-15-00368-f005:**
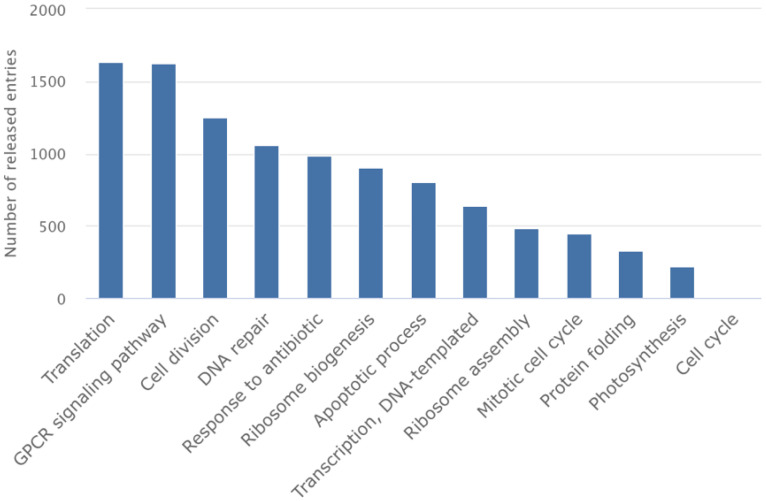
The importance of GPCRs. Number of EMDB entries revealing the high impact of GPCR research in structure-based drug discovery (accessed on November 2025).

**Table 1 membranes-15-00368-t001:** Comparison between Single Particle Analysis (SPA), Subtomogram Averaging (STA), Cryo-Electron Tomography (cryo-ET), and Micro-Electron Diffraction (microED).

Method	Sample Type	Key Strengths	Limitations	Typical Resolution	**Ideal Applications**
Single Particle Analysis (SPA)	Purified, isolated macromolecular complexes	Highest achievable resolution; no need for crystals; flexible conformational analysis	Requires many particles and biochemical purity; not ideal for heterogeneous cellular environments	~2–3 Å	Membrane proteins, complexes, dynamic assemblies in vitro
Subtomogram Averaging (STA)	Repeating structures inside cells or native environments (from tomograms)	Studies proteins in situ; preserves cellular context; tolerates heterogeneity	Lower particle numbers; limited by tomogram SNR; typically, lower resolution than SPA	~4–8 Å	Viral spikes, ribosomes in cells, large assemblies in native membranes
Cryo-Electron Tomography (cryo-ET)	Intact cells, organelles, thick specimens, pleomorphic structures	True 3D snapshots of cells; captures rare, heterogeneous states; no need for averaging	Lower resolution; missing wedge; high dose limits; requires thinning for thick samples	~3–5 nm	Cellular architecture, organelles, ultrastructure, native-state complexes
Micro-electron Diffraction (MicroED)	Nanocrystals of proteins or small molecules	Atomic resolution from crystals too small for X-ray; minimal sample quantity	Requires crystallization; data collection geometry is specialized	~1–2 Å	Small proteins, peptides, small molecules, membrane protein nanocrystals

## Data Availability

No new data were created or analyzed in this review. Data sharing does not apply to this article.

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
