# Peer review of "Biological Breakthroughs and Drug Discovery Revolution via Cryo-Electron Microscopy of Membrane Proteins"

_membranes, 2025, doi:10.3390/membranes15120368_

Round 1

Reviewer 1 Report

Comments and Suggestions for Authors

The paper describes the current advances in cryo-EM and how that has aided the study of membrane proteins involved in drug discovery. The background, procedures, limitations, and overall summary for each technique involved in cryo-EM have been described well. The manuscript also explains how the newly elucidated structures of proteins, such as ABC transporters, G-protein-coupled receptors, viral glycoproteins, the nuclear pore complex, and bacteriorhodopsin, have impacted our general understanding of cellular mechanisms. Lastly, there is a description of the challenges and strategies employed to overcome them, while keeping a positive outlook towards the future. The draft is very well written and easy to read. I’ll be pleased to recommend publication of this review.

Minor corrections:

Break up Figure 2 into panels. Keep legend brief. SPA needs to be mentioned and all steps labeled in the figure

Figure 2: The inset depicting the cryo-em grids has not been described in the legend and seems out of place here.

Figure 3 distinguishes tomography and sub-tomogram averaging as distinct slices of the pie-chart. A description of the difference between the two would help the reader.

Page 5, line 163 please replace similarly with similar

Figure 4: Please include the names of the proteins in the legend.

Author Response

I want to thank Reviewer's comments and all the suggestions were properly incorporated in the manuscript.

Reviewer 2 Report

Comments and Suggestions for Authors

The submitted review provides a broad overview of the role of cryo-electron microscopy (cryo-EM) — including Single Particle Analysis (SPA), Subtomogram Averaging (STA), and MicroED — in advancing the structural biology of membrane proteins and accelerating drug discovery efforts. The author outlines key biological achievements enabled by cryo-EM, covering ion channels, GPCRs, ABC transporters, viral glycoproteins, respiratory chain complexes, and other membrane-embedded assemblies. The manuscript highlights the “resolution revolution,” improvements in image processing. The topic is timely and relevant, and the manuscript contains substantial useful information. However, despite its strengths, the review requires some revision to meet the standards of Membranes.

Comment 1: Sections contain repeated or overlapping content. It is suggested to reorganize the review with clearer subsections, and reduce redundancy.

Comment 2: It would be good to include a comparative table summarizing the advantages and limitations of SPA vs. STA vs. MicroED.

Comment 3: Some sections (especially SPA and STA) are overly detailed and technical, while others (e.g., drug-design applications) remain superficial. Emphasize biological and pharmacological implications.

Comment 4: Given the manuscript title, the discussion of structure-based drug design is unexpectedly limited. Missing elements include case studies that demonstrate how cryo-EM directly influenced clinical candidate development. It is mentioned in the abstract that it will be discussed in the manuscript (“The integration of cryo-EM with computational drug design has already yielded clinical candidates and approved therapeutics, marking a new era in membrane protein pharmacology” line 18-20), but it is missing in the text.

Comment 5: It is suggested that an abbreviation list can be added. It will make following the paper much easier.  Additionally, the Author should ensure that all abbreviations are explained in the text or are not repeated (e.g., LMNG at lines 98 and 103).

Comment 6: Figure 2 – The title before the figure is not necessary. But it is worth adding the abbreviations (STA, SPA, and MicroED) to the Figure.

Comment 7: The text lacks a reference to Figure 3.

Comment 8: Breakthrough cryo-EM structures are presented on Figure 4 and discussed in chapter 3.1. Unfortunately, the manuscript does not provide the PDB accession codes corresponding to the structures depicted in Figure 4, making it difficult for readers to identify which specific cryo-EM structures are being discussed in chapter 3.1.

Comment 9: Figure 4 captions should include all abbreviation presented on the Figure.

Comment 10: Some names are suggested to be unified in manuscripts. For example:

  • “SPA-cryoEM” (Figure 4), “cryo-EM SPA-” (line 218), “SPA cryo-EM” (line 229 and 245) and “SPA-cryo-EM” (line 269) .
  • Micro-ED (Figure 4 and line 356), MicroED (line 214 and 370)

Comment 11: It is suggested that the manuscript be improved to avoid editorial mistakes (line 20 (.:), 83 (extra space), 250 (extra space), 311 (no comma at the end of the sentence), 329 (extra space)).

Comments on the Quality of English Language

The manuscript requires some language editing by a fluent English speaker or a professional editing service. Improving the clarity and consistency of the English will significantly enhance the overall quality and impact of the review.

Author Response

I would like to thank the Reviewer for their valuable suggestions and comments. I have incorporated all of them into the revised draft I have resubmitted.